# AIM: Adversarial Inference by Matching Priors and Conditionals

## Abstract

Effective inference for a generative adversarial model remains an important and challenging problem. We propose a novel approach, Adversarial Inference by Matching priors and conditionals (AIM), which explicitly matches prior and conditional distributions in both data and code spaces, and puts a direct constraint on the dependency structure of the generative model. We derive an equivalent form of the prior and conditional matching objective that can be optimized efficiently without any parametric assumption on the data. We validate the effectiveness of AIM on the MNIST, CIFAR-10, and CelebA datasets by conducting quantitative and qualitative evaluations. Results demonstrate that AIM significantly improves both reconstruction and generation as compared to other adversarial inference models.

## 1 Introduction

Deep directed generative models like variational autoencoder (VAE) (Kingma and Welling, 2013; Rezende et al., 2014) and generative adversarial network (GAN) (Goodfellow et al., 2014) have been proved to be powerful for modeling complex high-dimensional distributions. While both VAE and GAN can learn to generate realistic images, their underlying mechanisms are fundamentally different. VAE maps the data into low-dimensional codes using an encoder, and then reconstructs the original data by a decoder. This allows it to perform both generation and inference. GAN, on the other hand, trains a generator and a discriminator adversarially. The generator learns to fool the discriminator by mapping low-dimensional noise vectors to the data space; at the same time, the discriminator evolves to detect the generated fake samples from the true ones. These two methods have complementary strengths and weaknesses. VAE can learn a bidirectional mapping between data and code spaces, but relies on over-simplified parametric assumptions on the complex data distribution, thereby causing it to generate blurry images (Donahue et al., 2016; Goodfellow et al., 2014; Larsen et al., 2015). GAN generates more realistic samples than VAE (Radford et al., 2015; Larsen et al., 2015) because the adversarial regime allows it to learn more complex distributions. However, note that GAN only learns a unidirectional mapping for data generation, and does not allow inferring the latent codes from given samples. This is limiting because the ability of inference is very crucial for several downstream applications, such as classification, clustering, similarity search, and interpretation. Furthermore, GAN also suffers from the model collapse problem (Che et al., 2016; Salimans et al., 2016) – many modes of the data distribution are not represented in the generated samples.

Therefore, one may wonder on whether it is possible to develop a generative model that enjoys the strengths of both GAN and VAE without their inherent weaknesses. Such model should be able to generate high-quality samples as good as GAN, have an inference mechanism as effective as VAE, and also avoid the mode collapse issue. Many recent efforts have been devoted to combining VAE with adversarial discriminator(s) (Brock et al., 2016; Che et al., 2016; Larsen et al., 2015; Makhzani et al., 2015; Mescheder et al., 2017). However, VAE-GAN hybrids tend to manifest a compromise of the strengths and weaknesses of both the approaches. The main reason is that all of them retain the autoencoder structure, which requires an explicit metric to measure the data reconstruction and assumes over-simplified parametric data distributions. To overcome such limitations, adversarially learned inference (ALI) (Donahue et al., 2016; Dumoulin et al., 2016) was recently proposed, wherein the discriminator is trained on the joint distribution of data and latent codes. In this way, under a perfect discriminator, one can successfully match joint distributions of the decoder and encoder, thereby, performing inference by sampling from the encoder's conditional distribution that also matches the decoder's posterior. However, in practice the equilibrium of the jointly adversarial game

is hard to approach as the dependency structure between data and codes is not explicitly specified. ALI's reconstructions of samples are thus not always faithful (Dumoulin et al., 2016; Li et al., 2017) implying that its inference mechanism is not always effective.

To overcome the aforementioned issues, in this paper, we propose a novel approach, Adversarial Inference by Matching priors and conditionals (AIM), that integrates efficient inference to GAN and overcomes the limitations of prior approaches. The approach keeps the structure simple, involving only one generator, one encoder, and one discriminator. Furthermore, AIM's objective is directly derived from our goal of matching both prior and conditional distributions of the generator and encoder, instead of a heuristic combination with $l_k$ norm regularization. Compared to regular GANs, AIM has the ability to conduct inference, and also does not suffer from the mode collapse problem. Moreover, AIM also abandons the unrealistic parametric assumption on the conditional data distribution, and does not require any reconstruction in the data space. This is fundamentally different from VAE or VAE-GAN hybrids in which the $l_k$ norm is used to measure the data reconstruction. The usage of simple data-fitting metrics on the complex data distribution leads to worse generation performance. Different from ALI, AIM decomposes the hard problem of matching the joint distributions into two sub-tasks – explicitly matching the priors on the latent codes and the conditionals on the data. As a consequence of more restrictive constraint, it achieves better generation and more faithful reconstruction than ALI. Last but not the least, note that GAN variations with inference mechanism usually achieve worse generation performance as compared to regular GANs. However, as demonstrated in the experiments, AIM not only performs best on inference, but also improves the generation performance over GAN.

## 2 RELATED WORK

The most straightforward way to learn an inference mechanism is to learn the inverse mapping of GAN's generator post-hoc (Zhu et al., 2016). However, since its training process is the same as GAN, it still suffers from mode collapse problem. AGE (Ulyanov et al., 2017) encourages encoder and generator to be reciprocal by simultaneously minimizing an $l_1$ reconstruction error in the data space and an $l_2$ error in the code space. This is closely related to the cycle-consistency criterion (Zhu et al., 2017; Kim et al., 2017; Yi et al., 2017; Li et al., 2017). Although the pairwise reconstruction errors help reduce mode collapse, the data reconstruction is still measured by $l_1$ or $l_2$ norm, which brings the same problem of VAE and VAE-GAN hybrids. InfoGAN (Chen et al., 2016) minimizes the mutual information between a subset $\mathbf{c}$ of the latent code and the generated samples, and hence can only do partial inference on $\mathbf{c}$.

Different from the heuristic combination of VAE and GANs, Mescheder et al. (2017) theoretically derived an adversarial game to replace the KL-divergence term in the variational lower bound (also called ELBO), and gives the new method, adversarial variational Bayes (AVB), much more flexibility in its dependence on latent $\mathbf{z}$. However, the reconstruction term on $\mathbf{x}$ still exists and so is the parametric assumption on the conditional data distribution, leading to the blurriness in their reconstructed and generated samples.

ALI (Dumoulin et al., 2016; Donahue et al., 2016) is an elegant approach to bring inference mechanism into adversarial learning without assuming parametric distribution on the data. Different from our work, it directly plays an adversarial game to match the joint distributions of the decoder and encoder. But in practice, ALI's reconstructions are not necessarily faithful because the dependency structures within the two joint distributions are not specified (Li et al., 2017). ALICE (Li et al., 2017) tries to solve this problem by regularizing ALI using an extra conditional entropy constraint on the data. The conditional entropy is either explicitly measured by $l_k$ norm, or implicitly learned by adversarial training. However, when the data distribution becomes complicated (e.g. CIFAR-10), the $l_k$ metric may lead to blurry reconstructions and the adversarial training is hard to achieve (Li et al., 2017). Compared with ALI and ALICE, our method is proven to minimize the KL-divergence between both priors and conditionals of generator and encoder, and can provide consistent effective inference even on complicated distribution (see Section 5.1).

Srivastava et al. (2017) proposed VEEGAN to tackle the mode collapse issue of GANs by adding an implicit variatinoal learning on the latent $\mathbf{z}$. To our best knowledge, this is by far the only approach that is also reconstructing $\mathbf{z}$. Different from VAEs, VEEGAN autoencodes the latent variable or noise $\mathbf{z}$. By doing so, it enforces the generator not to collapse the mappings of $\mathbf{z}$ to a single mode, because

otherwise, the encoder will not be able to recover all the noise $\mathbf{z}$. The details of their model can be summarized as ALI regularized by an extra reconstruction of latent $\mathbf{z}$. Therefore, VEEGAN is similar to ALICE in the sense that they are both adversarial games on the joint distribution with an extra regularization on either data or latent reconstruction. Our model AIM instead only plays the adversarial game on the marginal data distribution, and reconstructs the latent $\mathbf{z}$ by maximizing its log-likelihood under the latent posterior distribution.

## 3 BACKGROUND

We consider the generative model $p_\theta(\mathbf{z})p_\theta(\mathbf{x}|\mathbf{z})$, where a latent variable $\mathbf{z}^{(i)}$ is first generated from the prior distribution $p_\theta(\mathbf{z})$, and then the data $\mathbf{x}^{(i)}$ is sampled from the conditional distribution $p_\theta(\mathbf{x}|\mathbf{z})$. The parameter $\theta$ stands for the ground truth parameter of the distribution. The prior $p_\theta(\mathbf{z})$ is always assumed to be a simple parametric distribution (e.g. $\mathcal{N}(\mathbf{0}, \mathbf{I})$), but the generative conditional $p_\theta(\mathbf{x}|\mathbf{z})$ is always very complex and not known to us.

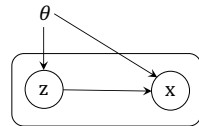

Figure 1: This graphical model structure is shared with GAN and VAE.

Moreover, the posterior $p_\theta(\mathbf{z}|\mathbf{x})$ is intractable but stands for an important inference procedure: given a data $\mathbf{x}^{(i)}$, it allows us to infer its latent variable $\mathbf{z}^{(i)}$.

## 4 METHODOLOGY

In our method, we will model the generating process by a neural network called *generator*, and the inference process by another neural network *encoder*. Consider the following two distributions of the generator and encoder, and their corresponding sampling procedures:

- the *generator* distribution: $p(\mathbf{z})p(\mathbf{x}|\mathbf{z})$; $\mathbf{z} \sim p(\mathbf{z})$, $\mathbf{x} \sim p(\mathbf{x}|\mathbf{z})$.
- the *encoder* distribution: $q(\mathbf{x})q(\mathbf{z}|\mathbf{x})$; $\mathbf{x} \sim q(\mathbf{x})$, $\mathbf{z} \sim q(\mathbf{z}|\mathbf{x})$.

The generator's conditional $p(\mathbf{x}|\mathbf{z})$ approximates the generating distribution $p_\theta(\mathbf{x}|\mathbf{z})$. The encoder's conditional $q(\mathbf{z}|\mathbf{x})$ approximates the posterior distribution $p_\theta(\mathbf{z}|\mathbf{x})$, which is what we need for inference. The marginal distribution $q(\mathbf{x})$ stands for the empirical data distribution, and the other marginal $p(\mathbf{z})$ is taken to be $p_\theta(\mathbf{z})$, which is always a known distribution like standard Gaussian.

### 4.1 MATCHING PRIORS AND CONDITIONALS

The ultimate goal is to match the joint distributions, $p(\mathbf{x}, \mathbf{z})$ and $q(\mathbf{x}, \mathbf{z})$. If this is achieved, we are guaranteed that all marginals match and all conditionals match as well. In particular, the conditional $q(\mathbf{z}|\mathbf{x})$ matches the posterior $p(\mathbf{z}|\mathbf{x})$. We propose to decompose this goal into two sub-tasks – matching the priors $p(\mathbf{z})$ and $q(\mathbf{z})$, and matching the conditionals $p(\mathbf{x}|\mathbf{z})$ and $q(\mathbf{x}|\mathbf{z})$. There are two advantages. Firstly, we explicitly define the dependency structure $\mathbf{z} \rightarrow \mathbf{x}$. Secondly, the explicit constraints on both priors and conditionals are stronger than merely one constraint on the joint distributions.

In order to match the distributions, we will minimize the KL-divergence between the priors and between the conditionals:

$$\mathbb{E}_{p(\mathbf{z})} D_{KL}(p(\mathbf{x}|\mathbf{z})||q(\mathbf{x}|\mathbf{z})) + D_{KL}(p(\mathbf{z})||q(\mathbf{z})). \tag{1}$$

By the properties of KL-divergence, when the minimum of equation 1 is attained, we have both $p(\mathbf{z}) = q(\mathbf{z})$ and $p(\mathbf{x}|\mathbf{z}) = q(\mathbf{x}|\mathbf{z})$, for all $\mathbf{x}$ and $\mathbf{z}$.

### 4.2 OBJECTIVE FUNCTION

The objective equation 1 cannot be directly optimized because both $q(\mathbf{z})$ and $q(\mathbf{x}|\mathbf{z})$ are impossible to sample from, as the flow in the encoder is from $\mathbf{x}$ to $\mathbf{z}$. Therefore, we rewrite equation 1 as the combination of a KL-divergence term and a reconstruction term, so that both of them only contain distributions that can be sampled from or directly evaluated.

Firstly, for any fixed $\mathbf{z}$, the KL-divergence between the conditionals, $D_{KL}(p(\mathbf{x}|\mathbf{z})||q(\mathbf{x}|\mathbf{z}))$, can be decomposed as:

$$
\begin{aligned}
D_{KL}(p(\mathbf{x}|\mathbf{z})||q(\mathbf{x}|\mathbf{z})) &= \mathbb{E}_{p(\mathbf{x}|\mathbf{z})}\left[\log p(\mathbf{x}|\mathbf{z}) - \log q(\mathbf{x}) - \log q(\mathbf{z}|\mathbf{x}) + \log q(\mathbf{z})\right] \\
&= D_{KL}(p(\mathbf{x}|\mathbf{z})||q(\mathbf{x})) - \mathbb{E}_{p(\mathbf{x}|\mathbf{z})}[\log q(\mathbf{z}|\mathbf{x})] + \log q(\mathbf{z}). \quad (2)
\end{aligned}
$$

Then by adding $(\log p(\mathbf{z}) - \log p(\mathbf{z})) - \log q(\mathbf{z})$ to both sides of equation 2, we have

$$
D_{KL}(p(\mathbf{x}|\mathbf{z})||q(\mathbf{x}|\mathbf{z})) + (\log p(\mathbf{z}) - \log q(\mathbf{z})) - \log p(\mathbf{z}) = D_{KL}(p(\mathbf{x}|\mathbf{z})||q(\mathbf{x})) - \mathbb{E}_{p(\mathbf{x}|\mathbf{z})}[\log q(\mathbf{z}|\mathbf{x})].
$$

Now taking expectation with respect to $p(\mathbf{z})$ on both sides, we get

$$
\begin{aligned}
&\mathbb{E}_{p(\mathbf{z})} D_{KL}(p(\mathbf{x}|\mathbf{z})||q(\mathbf{x}|\mathbf{z})) + D_{KL}(p(\mathbf{z})||q(\mathbf{z})) - \mathbb{E}_{p(\mathbf{z})}\log p(\mathbf{z}) \\
&= \underbrace{\mathbb{E}_{p(\mathbf{z})} D_{KL}(p(\mathbf{x}|\mathbf{z})||q(\mathbf{x}))}_{\text{I}} + \underbrace{\mathbb{E}_{p(\mathbf{z})}\mathbb{E}_{p(\mathbf{x}|\mathbf{z})}[-\log q(\mathbf{z}|\mathbf{x})]}_{\text{II}}. \quad (3)
\end{aligned}
$$

Note that the term $\mathbb{E}_{p(\mathbf{z})}\log p(\mathbf{z})$ is a constant because the prior $p(\mathbf{z})$ is a fixed parametric distribution. For example, when $\mathbf{z} \sim \mathcal{N}(\mathbf{0}, \mathbf{I}_d)$, we have $\mathbb{E}_{p(\mathbf{z})}\log p(\mathbf{z}) = -d(1 + \log(2\pi))/2$. Therefore, minimizing the objective equation 1 is now transformed to minimizing the new objective equation 3. Intuitively, term (I) measures the difference between generated and real samples, and term (II) measures the reconstruction of the latent codes. We summarize the above procedure as a proposition.

**Proposition 1** *The objective function*

$$
\mathbb{E}_{p(\mathbf{z})}\left\{D_{KL}(p(\mathbf{x}|\mathbf{z})||q(\mathbf{x})) + \mathbb{E}_{p(\mathbf{x}|\mathbf{z})}[-\log q(\mathbf{z}|\mathbf{x})]\right\}
$$

*is minimized when $p(\mathbf{z}) = q(\mathbf{z})$ and $p(\mathbf{x}|\mathbf{z}) = q(\mathbf{x}|\mathbf{z})$ for all $\mathbf{z}$, $\mathbf{x}$. And hence, at the minimum, the joint distributions $p(\mathbf{x}, \mathbf{z}) = q(\mathbf{x}, \mathbf{z})$.*

## 4.3 RELATION TO VAE

The VAE (Kingma and Welling, 2013) method, using our notations in this paper, actually depends on the following identity:

$$
D_{KL}(q(\mathbf{z}|\mathbf{x})||p(\mathbf{z}|\mathbf{x})) = D_{KL}(q(\mathbf{z}|\mathbf{x})||p(\mathbf{x})) - \mathbb{E}_{q(\mathbf{z}|\mathbf{x})}[\log p(\mathbf{x}|\mathbf{z})] + \log p(\mathbf{x}). \quad (4)
$$

Then because of the non-negativity of KL-divergence, we have

$$
\log p(\mathbf{x}) \geq \underbrace{\mathbb{E}_{q(\mathbf{z}|\mathbf{x})}[\log p(\mathbf{x}|\mathbf{z})] - D_{KL}(q(\mathbf{z}|\mathbf{x})||p(\mathbf{x}))}_{\text{ELBO}},
$$

and hence maximizing the log-likelihood of the observations can be transferred to maximizing the evidence lower bound (ELBO). But taking a closer look at equation 4 and comparing it to equation 2, we notice that equation 4 is a decomposition of the KL-divergence between two conditionals, $q(\mathbf{z}|\mathbf{x})$ and $p(\mathbf{z}|\mathbf{x})$. Therefore, we can follow the same approach after equation 2 and get the following identity:

$$
\begin{aligned}
&\mathbb{E}_{q(\mathbf{x})} D_{KL}(q(\mathbf{z}||\mathbf{x})||p(\mathbf{z}|\mathbf{x})) + D_{KL}(q(\mathbf{x})||p(\mathbf{x})) - \mathbb{E}_{q(\mathbf{x})}\log q(\mathbf{x}) \\
&= \underbrace{\mathbb{E}_{q(\mathbf{x})} D_{KL}(q(\mathbf{z}|\mathbf{x})||p(\mathbf{z}))}_{\text{I}_{vae}} + \underbrace{\mathbb{E}_{q(\mathbf{x})}\mathbb{E}_{q(\mathbf{z}|\mathbf{x})}[-\log p(\mathbf{x}|\mathbf{z})]}_{\text{II}_{vae}}. \quad (5)
\end{aligned}
$$

Since the marginal $q(\mathbf{x})$ stands for the empirical data distribution, the right hand side of equation 5 is the empirical expectation of the negative ELBO, which is what VAE tries to minimize. We then conclude from equation 5 that VAE performs marginal distribution matching in the data space and conditional distribution matching in the latent space. This distribution matching of VAE is also observed by Rosca et al. (2018).

However, the marginal distributions in the data space are very complex, and the direction $\mathbf{x} \to \mathbf{z}$ in the conditional distributions in the latent space is actually opposite to the generating process $\mathbf{z} \to \mathbf{x}$. Hence, in order to match these distributions, VAE's objective has a reconstruction term $\text{II}_{vae}$ on $\mathbf{x}$, and a regularization term $\text{I}_{vae}$ on latent $\mathbf{z}$. But to evaluate both terms, we need to make parametric assumptions on both conditionals $q(\mathbf{z}|\mathbf{x})$ and $p(\mathbf{x}|\mathbf{z})$. The assumption on $\text{I}_{vae}$ can be loosed using

GANs (Makhzani et al., 2015), but the assumption on $\mathrm{II}_{vae}$ is critical and limits the performance of VAE-GAN hybrids.

Our model, AIM, instead performs marginal distribution matching in the latent space and conditional distribution matching in the data space. From equation 3, since the term I will be replaced with an adversarial game (see Section 4.4), the only assumption we need to make is on term II, that is, on the conditional $q(\mathbf{z}|\mathbf{x})$. And our model is very flexible in its dependence on $\mathbf{z}$. This assumption is much weaker than that on $p(\mathbf{x}|\mathbf{z})$ and does not lead to the problems of VAE or VAE-GANs (e.g. blurriness).

## 4.4 AIM FRAMEWORK

The KL-divergence part (I) can be replaced by an adversarial game using the $f$-divergence theory (Nowozin et al., 2016). The reconstruction term (II) is a log-likelihood and can be simply evaluated if we assume a parametric $q(\mathbf{z}|\mathbf{x})$. Therefore, our framework only requires exactly one *generator G*, one *discriminator D*, and one *encoder E*. We will now discuss how to play the adversarial game and measure the reconstruction in details.

**Adversarial game**    Because we do not want to make any parametric assumption on the distribution $p(\mathbf{x}|\mathbf{z})$, an adversarial game will be played to distinguish $p(\mathbf{x}|\mathbf{z})$ from $q(\mathbf{x})$. By the theory of $f$-GAN (Nowozin et al., 2016), we then construct an adversarial game with the value function

$$V(G, D) = \mathbb{E}_{\mathbf{x} \sim q(\mathbf{x})}[D(\mathbf{x})] + \mathbb{E}_{\mathbf{z} \sim p(\mathbf{z})}[-\exp(D(G(\mathbf{z})) - 1)]. \tag{6}$$

Under the perfect discriminator, finding the optimal generator is then equivalent to minimizing the KL-divergence. The activation function for the discriminator in equation 6 is just the identity mapping instead of the sigmoid function in the original GAN. But just like in the original GAN, the generator of equation 6 also suffers from the gradient vanishing problem (Goodfellow, 2016). Since the original GAN is well studied and suffices for our purpose, we will instead use its value function:

$$V(G, D) = \mathbb{E}_{\mathbf{x} \sim q(\mathbf{x})}[\log(D(\mathbf{x}))] + \mathbb{E}_{\mathbf{z} \sim p(\mathbf{z})}[\log(1 - D(G(\mathbf{z})))]. \tag{7}$$

And as suggested in Goodfellow (2016), in order to mitigate the generator's gradient vanishing problem, we minimize another value function $-\mathbb{E}_{\mathbf{z} \sim p(\mathbf{z})}[\log D(G(\mathbf{z}))]$ for the generator.

**Reconstruction**    Because of the simplicity of the distribution on $\mathbf{z}$, we make a reasonable parametric assumption on $q(\mathbf{z}|\mathbf{x})$ so that the log-likelihood can be explicitly calculated. In this paper we will assume $\mathbf{z}|\mathbf{x} \sim \mathcal{N}(\mu(\mathbf{x}), \sigma^2(\mathbf{x})\mathbf{I})$, and hence

$$-\log q(\mathbf{z}|\mathbf{x}) = \frac{1}{2} \sum_{j=1}^{d} \left( \frac{(z_j - \mu_j(\mathbf{x}))^2}{\sigma_j^2(\mathbf{x})} + \log \sigma_j^2(\mathbf{x}) + \log(2\pi) \right) =: L(\mathbf{z}, \mu(\mathbf{x}), \sigma^2(\mathbf{x})), \tag{8}$$

where $d$ is the dimension of $\mathbf{z}$. In this case, the encoder network only needs to output two vectors, $\mu(\mathbf{x})$ and $\sigma^2(\mathbf{x})$, that is, $E(\mathbf{x}) = (\mu(\mathbf{x}), \sigma^2(\mathbf{x}))$. Then we can compute the approximate negative posterior log-likelihood by plugging $E(\mathbf{x})$ into equation 8, i.e. $L(\mathbf{z}, E(\mathbf{x}))$.

To summarize, our final optimization problem is

$$\min_{G, E} \max_{D} \Big\{ V(G, D) + \lambda \mathbb{E}_{p(\mathbf{z})}[L(\mathbf{z}, E(G(\mathbf{z})))] \Big\}. \tag{9}$$

Here, if we use $V(G, D)$ in equation 6 derived from $f$-GAN, then $\lambda = 1$. If we use $V(G, D)$ in equation 7, then $\lambda$ needs to be set so that two parts of equation 9 are in the same scale. We will discuss this in detail in Section 5.

## 4.5 TRAINING AND INFERENCE PROCEDURES

The training procedure is summarized in Algorithm 1. Given random $\mathbf{z}^{(i)} \sim p(\mathbf{z})$, we first generate samples $\tilde{\mathbf{x}}^{(i)} \sim p(\mathbf{x}|\mathbf{z}^{(i)})$ using the generator. Then the discriminator is updated to distinguish between generated and real samples. The encoder outputs the parameters for the distribution $q(\mathbf{z}|\mathbf{x})$, from which we calculate the log-likelihood in (II). Then the generator and encoder are updated together to minimize the reconstruction error (i.e. maximize the expected log-likelihood), while the generator has an extra goal that is to fool the discriminator. The inference procedure is in Algorithm

2. For any data $\mathbf{x}^{(i)}$, its inferred latent code is set to be the conditional mean $\mu(\mathbf{x}^{(i)}) = \mathbb{E}_{q(\mathbf{z}|\mathbf{x}^{(i)})}[\mathbf{z}]$. Then the reconstruction of $\mathbf{x}^{(i)}$ is $G(\mu(\mathbf{x}^{(i)}))$. Besides the reconstruction, we can also generate more samples which are close to $\mathbf{x}^{(i)}$ in the sense that they have similar latent codes. This can be done by first sampling $\mathbf{z}$'s from the posterior $q(\mathbf{z}|\mathbf{x}^{(i)})$, and then map them to the data space using the generator.

---

**Algorithm 1** The AIM training procedure.

---
$\theta_g, \theta_d, \theta_e \leftarrow$ initialize network parameters
**repeat**
    $\mathbf{z}^{(1)}, \ldots, \mathbf{z}^{(n)} \sim p(\mathbf{z})$          ▷ Draw $n$ samples from the prior
    $\tilde{\mathbf{x}}^{(j)} \leftarrow G(\mathbf{z}^{(j)}), \quad j = 1, \ldots, n$     ▷ Generate samples using the generator network
    $(\mu(\tilde{\mathbf{x}}^{(j)}), \sigma^2(\tilde{\mathbf{x}}^{(j)})) \leftarrow E(\tilde{\mathbf{x}}^{(j)})$     ▷ Calculate mean and variance of $q(\mathbf{z}|\tilde{\mathbf{x}}^{(j)})$
    $\rho_q^{(i)} \leftarrow D(\mathbf{x}^{(i)}), \quad i = 1, \ldots, n$      ▷ Compute discriminator predictions
    $\rho_p^{(j)} \leftarrow D(\tilde{\mathbf{x}}^{(j)}), \quad j = 1, \ldots, n$
    $\mathcal{L}_d \leftarrow -\frac{1}{n}\sum_{i=1}^{n} \log(\rho_q^{(i)}) - \frac{1}{n}\sum_{j=1}^{n} log(1 - \rho_p^{(j)})$    ▷ Compute discriminator loss
    $\mathcal{L}_g \leftarrow -\frac{1}{n}\sum_{j=1}^{n} \log(\rho_p^{(j)})$       ▷ Compute generator loss
    $\mathcal{L}_e \leftarrow \frac{\lambda}{n}\sum_{j=1}^{n} L(\mathbf{z}^{(j)}, \mu(\tilde{\mathbf{x}}^{(j)}), \sigma^2(\tilde{\mathbf{x}}^{(j)}))$     ▷ Compute encoder loss
    $\mathcal{L}_{rec} \leftarrow \mathcal{L}_g + \mathcal{L}_e$        ▷ Compute reconstruction loss
    $\theta_d \leftarrow \theta_d - \nabla_{\theta_d}\mathcal{L}_d$      ▷ Gradient update on discriminator network
    $(\theta_g, \theta_e) \leftarrow (\theta_g, \theta_e) - \nabla_{(\theta_g, \theta_e)}\mathcal{L}_{rec}$    ▷ Gradient update on generator and encoder networks
**until** convergence

---

**Algorithm 2** The AIM inference procedure.

---
$\mathbf{x}^{(i)} \sim q(\mathbf{x})$              ▷ Data to do inference on
$(\mu(\mathbf{x}^{(i)}), \sigma^2(\mathbf{x}^{(i)})) \leftarrow E(\mathbf{x}^{(i)})$      ▷ Calculate mean and variance of $q(\mathbf{z}|\mathbf{x}^{(i)})$
$\tilde{\mathbf{x}}^{(i)} \leftarrow G(\mu(\mathbf{x}^{(i)}))$         ▷ Get reconstruction of $\mathbf{x}^{(i)}$
$\mathbf{z}^{(j)} \sim \mathcal{N}(\mu(\mathbf{x}^{(i)}), \sigma^2(\mathbf{x}^{(i)})), \quad j = 1, \ldots, m$   ▷ Sample $\mathbf{z}$ from the posterior $q(\mathbf{z}|\mathbf{x}^{(i)})$
$\tilde{\mathbf{x}}^{(i,j)} \leftarrow G(\mathbf{z}^{(j)}), \quad j = 1, \ldots, m$     ▷ Get similar neighbors of $\mathbf{x}^{(i)}$

---

### 4.6 REPARAMETRIZATION

In order to sample from $p(\mathbf{x}|\mathbf{z})$ and $q(\mathbf{z}|\mathbf{x})$, we use the reparametrization trick (Kingma and Welling, 2013; Bengio et al., 2013; 2014). Instead of directly sampling from a complex distribution, we can reparametrize the random variable as a deterministic transformation of an auxiliary noise variable $\epsilon$, that is, $u = f(v, \epsilon)$. In our case, to sample from $\tilde{\mathbf{x}} \sim p(\mathbf{x}|\mathbf{z})$, we can rewrite $\tilde{\mathbf{x}} = G(\mathbf{z}, \epsilon)$, where $\mathbf{z} \sim p(\mathbf{z})$ and $\epsilon$ is an auxiliary variable from a simple distribution. If we do not pass in any auxiliary noise $\epsilon$, then the generator will be deterministic. For the encoder part, since $q(\mathbf{z}|\mathbf{x}) = \mathcal{N}(\mu(\mathbf{x}), \sigma^2(\mathbf{x})\mathbf{I})$, one can draw samples by computing $\mathbf{z} = \mu(\mathbf{x}) + \sigma(\mathbf{x}) \odot \epsilon$, where $\epsilon \sim \mathcal{N}(\mathbf{0}, \mathbf{I})$.

## 5 EXPERIMENTS

We evaluate our proposed method, AIM, for both reconstruction and generation tasks, on the data sets MNIST (LeCun et al., 1998), CIFAR-10 (Krizhevsky and Hinton, 2009) and CelebA (Liu et al., 2015). To show insights behind the effectiveness of AIM, we also conduct the same 2D Gaussian mixture experiment as in Dumoulin et al. (2016), and explore the 2D latent representation on MNIST. The architectures of our discriminator and generator are based on DCGAN (Radford et al., 2015) and slightly simpler, which can be easily replaced by more advanced state-of-the-art GANs, and we use a deterministic generator throughout the experiments. Our encoder network consists of convolutional layers followed by two separated fully connected networks, which are used to predict the mean and variance of the posterior $q(\mathbf{z}|\mathbf{x})$, respectively. The Adam optimizer (Kingma and Ba, 2014) is used and the learning rate decay strategy suggested by Kingma and Ba (2014) is applied. Since the input to the log-function is one-dimensional in equation 7 but $d$-dimensional in equation 8, we set $\lambda$ to be $1/d$. We also observe that the discriminator shares a similar task with the encoder: both of them need to

extract higher level features from raw images. Therefore, in order to reduce the number of parameters and to stabilize the training procedure, our encoder takes the intermediate hidden representation learned by the discriminator as its own input. It is worth noting that the encoder does not update the common feature extracting layers. More details about the model architecture and optimization methods are listed in the appendices.

## 5.1 QUANTITATIVE RESULTS ON REAL DATASETS

In this section, we use quantitative measures to compare the inference and generation performance of AIM, GAN, ALI and ALICE. And for fair comparison, GAN is implemented to have the identical generator and discriminator with AIM. We also include a reduced version of AIM, named AIM−, in which the conditional distribution $q(\mathbf{z}|\mathbf{x})$ of the encoder is assumed to be a Gaussian with identity covariance matrix. To evaluate the performance of inference, we measure it through reconstructing test images and calculating the mean squared error (MSE), which has been adopted in Li et al. (2017). As for generation, we calculate the inception score (Salimans et al., 2016) on $50,000$ randomly generated images. The inception scores on MNIST are evaluated by the pre-trained classifier from Li et al. (2017), and the inception scores on CIFAR-10 is based on the ImageNet. The results are summarized in Table 1.

Table 1: MSE (lower is better) and Inception scores (higher is better) on MNIST and CIFAR-10. ALI and ALICE results are from the experiments in Li et al. (2017).

| Method | MNIST | | CIFAR-10 | |
|---|---|---|---|---|
| | MSE | Inception Score | MSE | Inception Score |
| GAN | – | $9.464 \pm 0.020$ | – | $6.287 \pm 0.061$ |
| ALI | $0.480 \pm 0.100$ | $8.749 \pm 0.090$ | $0.672 \pm 0.113$ | $5.930 \pm 0.044$ |
| ALICE | $0.080 \pm 0.007$ | $9.279 \pm 0.070$ | $0.416 \pm 0.202$ | $6.015 \pm 0.028$ |
| AIM− | $0.028 \pm 0.018$ | $9.331 \pm 0.021$ | $0.037 \pm 0.017$ | $6.324 \pm 0.056$ |
| AIM | $\mathbf{0.026 \pm 0.018}$ | $\mathbf{9.483 \pm 0.020}$ | $\mathbf{0.019 \pm 0.009}$ | $\mathbf{6.450 \pm 0.085}$ |

**Inference** From Table 1, AIM achieves the best reconstruction results on both data sets. On MNIST, AIM significantly decreases the MSE by 68% and 95% compared with ALICE and ALI respectively. On the more complicated CIFAR-10 data set, AIM decreases the MSE by 95% and 97%. In order to alleviate the non-identifiable issue of ALI, ALICE adds the conditional entropy constraint by explicitly regularizing the $l_k$ norms between the reconstructed and real images. However, as the data distribution becomes more complicated like in CIFAR-10, the $l_k$ norms become inadequate to measure the reconstruction. Consequently, ALICE's reconstruction error on CIFAR-10 increases significantly compared with that on MNIST. In contrast, the reconstruction performance of AIM is consistent on both data sets. The reason is that our model explicitly specifies the dependency structure of the generative model, and matches both prior and conditional distributions without using the simple data-fitting $l_k$ metrics in the data space. This can be further justified by the performance of AIM− which follows the same structure. Compared with AIM−, AIM further decreases the MSE significantly by 49% on CIFAR-10, which shows that the inferred conditional variance is crucial for achieving the faithful reconstructions on complicated data sets.

**Generation** AIM outperforms all the baseline models including GAN on inception score. This suggests that AIM can bring further improvement on generation performance instead of deteriorating it like the other baselines. The reason that both ALI and ALICE perform worse than GAN on generation is that the task of matching two complicated joint distributions, $p(\mathbf{z}, \mathbf{x})$ and $q(\mathbf{z}, \mathbf{x})$, is more difficult than the task of the regular GAN, which is to match only the marginals, $p(\mathbf{x})$ and $q(\mathbf{x})$. The proposed model AIM explicitly defines the dependency structure between $\mathbf{z}$ and $\mathbf{x}$, which is more effective compared with one step joint distribution matching. Comparison between AIM and AIM− shows that the learned variance is also critical for better generation performance. We also want to highlight that AIM's generation performance can be further improved by replacing the adversarial network with more advanced state-of-the-art GANs.

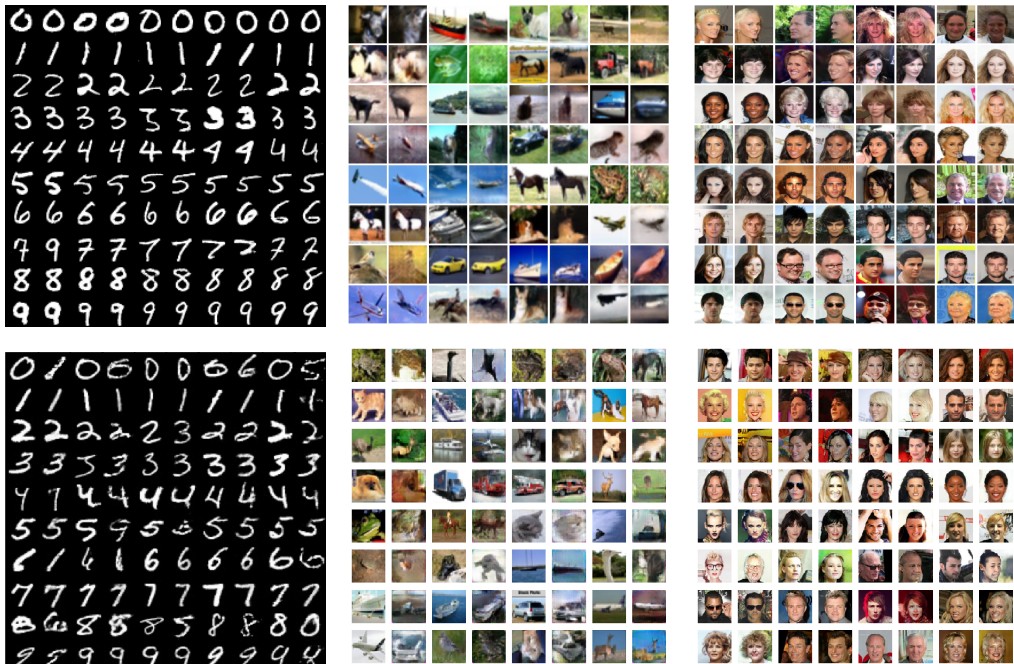

Figure 2: Reconstruction comparison between our proposed model AIM (first row) and ALI (BiGAN) (second row) on MNIST, CIFAR-10 and CelebA. In each of the figures, the odd columns are original samples from the test set and the even columns are their reconstructions.

## 5.2 RECONSTRUCTION RESULTS

We compare the reconstruction produced by AIM with the results of joint distribution regimes. Since ALICE is sensitive to hyper-parameter, we show its reconstruction results with different hyper-parameter settings in Appendix, from which we observe either unfaithful reconstructions or blurriness. In Figure 2, we compare reconstruction of AIM with the results reported in ALI(Dumoulin et al., 2016) (BiGAN (Donahue et al., 2016)). From the first column of Figure 2, we observe that ALI provides a certain level of reconstructions. However, it fails to capture the precise style of the original digits. In contrast, AIM can achieve very sharp and faithful reconstructions. On the more complicated dataset CIFAR-10, ALI's reconstructions are less faithful and oftentimes make mistakes in capturing exact object placement, color, style, and object identity. Our model produces better reconstructions in all these aspects. For the reconstructions of CelebA in column three, AIM reproduces the similar style, color and face placement, and even achieves a high level of face identity. As stated in Dumoulin et al. (2016), they believe ALI's unfaithful reconstructions is caused by underfitting. This also leads us to believe that our adversarial regime (marginal and conditional distribution matching) is more efficient for inference compared to joint distribution matching regimes.

## 5.3 MODE COLLAPSE REDUCTION

Table 2: Degree of mode collapse, measured by modes captured (higher is better) and % high quality samples (higher is better) on 2D grid data. The baseline results are reported in Srivastava et al. (2017).

|  | Vanilla GAN | ALI | Unrolled GAN | VEEGAN | AIM(ours) |
|---|---|---|---|---|---|
| Modes (Max 25) | 3.3 | 15.84 | 23.6 | 24.6 | **25** |
| % High Quality Samples | 0.5 | 1.6 | 16 | 40 | **81.1** |

To show the effectiveness of our model on mode collapse reduction, we perform the same synthetic experiment as in Dumoulin et al. (2016). The data is a 2D Gaussian mixture of 25 components laid out on a grid as shown in Figure 3a. To quantify the degree of mode collapse, we use the two metrics used in Srivastava et al. (2017): the *number of modes captured* and the *percentage of high quality samples*. A generated sample is counted as high quality if it is within three standard deviations of

the nearest mode. Then the number of modes captured is the number of mixture components whose mean is nearest to at least one high quality sample.

As shown in Table 2, the proposed model AIM outperforms the other state-of-the-art models consistently on both measures. More specifically, AIM can capture 25 modes every time and generate more than 80% of high-quality samples. This suggests that the proposed model AIM significantly alleviates the mode collapse issue of the GAN framework and hence further improves the generation performance.

To further demonstrate the insights behind our model, we show the inferred latent codes, reconstructions, and generated samples by AIM together with the results by ALI in Figure 3. Note that the ALI's result here is produced by the best-covering model in their paper, while the average number of modes captured by ALI is 15.84 compared to ours 25. From Figure 3b, we observe that our latent codes exhibit a disk shape and lie within the three standard deviations area of the standard Gaussian prior. Furthermore, they keep the relative spacial relationship of the test samples. In Figure 3c, AIM's reconstruction is almost identical to the original data (Figure 3a), while ALI's reconstruction is less faithful and consists of connecting points between modes. Both models have similar generating performance as shown in Figure 3d. Some generated samples lie in the regions between the true modes. This is because the true modes are separated by large low-probability regions. It does not align well with the assumption of continuous prior distribution.

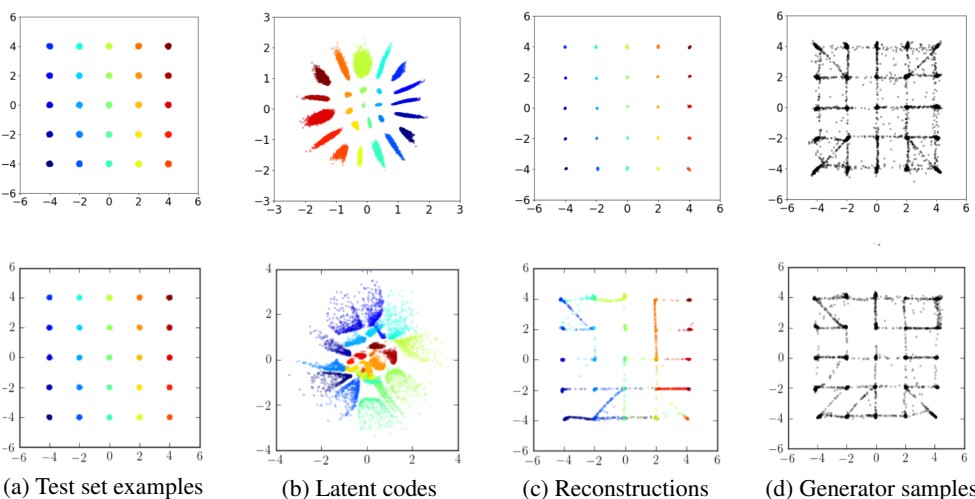

|  (a) Test set examples | (b) Latent codes | (c) Reconstructions | (d) Generator samples |

Figure 3: Comparison of our proposed model AIM (first row) with ALI (second row). The ALI's result shown here is the selected best-covering result reported in their original paper. And to follow the same settings with ALI, the number of points shown in column (b,c,d) is 10,000 which is 10% of the points shown in column (a).

## 5.4 LATENT REPRESENTATION OF MNIST

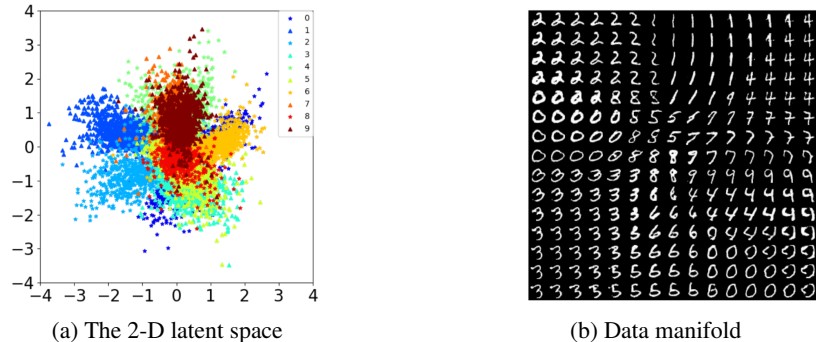

|  (a) The 2-D latent space | (b) Data manifold |

Figure 4: 2D latent codes of MNIST samples and the data manifold.

The 2D latent codes of MNIST test data inferred by AIM is shown in Figure 10a. In a totally unsupervised manner, AIM does a great job at clustering the similar digits. The latent codes still lie in the standard Gaussian range but now exhibit no "holes", compared to Figure 3b. This is because the data distribution of MNIST is much more continuous than the Gaussian mixture. In Figure 10c, we show images generated by linearly interpolating between $-2$ and $2$ along each dimension of the latent space. The generated images are sharp and the transitions between digits are smooth. This indicates that AIM learns smooth and meaningful representations which can generalize well.

## 5.5 UNSUPERVISED CONDITIONAL GENERATION

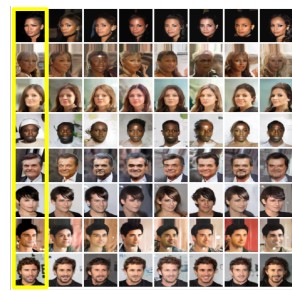

Because the explicit posterior distribution $q(\mathbf{z}|\mathbf{x})$ can be learned by the proposed model AIM, we can generate the similar samples conditioned on a given sample. This is meaningful in downstream task like data augmentation. In Figure 5, based on any image $\mathbf{x}^{(i)}$ in the first column, we conditionally generate more images by sampling $\mathbf{z}$ from the posterior $q(\mathbf{z}|\mathbf{x}^{(i)})$. The conditionally generated images are shown in the same row of $\mathbf{x}^{(i)}$. We observe that they have the similar style, color, and face placement with the original image.

Figure 5: Conditionally generated samples on the CelebA.

## 6 CONCLUSION

We proposed a novel framework, AIM, which matches both prior and conditional distributions between the generator and the encoder. Adversarial inference is incorporated into this framework and there is no parametric assumption on the conditional data distribution. Therefore, the proposed approach can not only learn an efficient inference mechanism but also improve the fidelity of generated samples. Extensive experiments on real datasets validate the effectiveness of the proposed model, and some insights are revealed by experiments on synthetic data.

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

# A    MORE EXPERIMENT RESULTS

## A.1    COMPARISON BETWEEN AIM'S AND ALICE'S RECONSRUCTIONS ON CELEBA

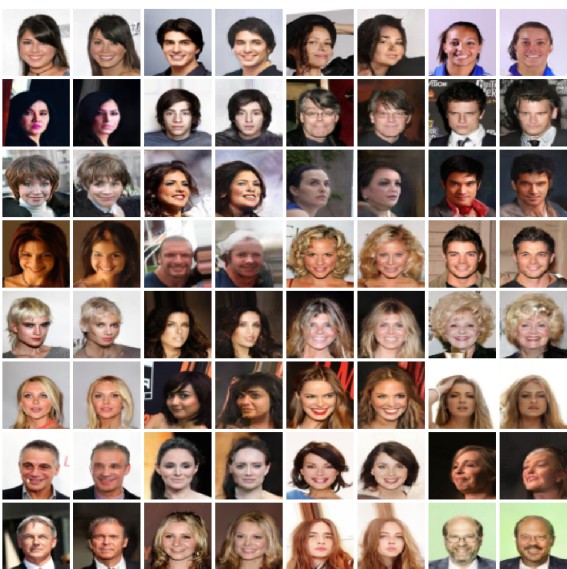

Figure 6: AIM's CelebA reconstructions. The odd columns are original samples from the CelebA test set and the even columns are their reconstructions.

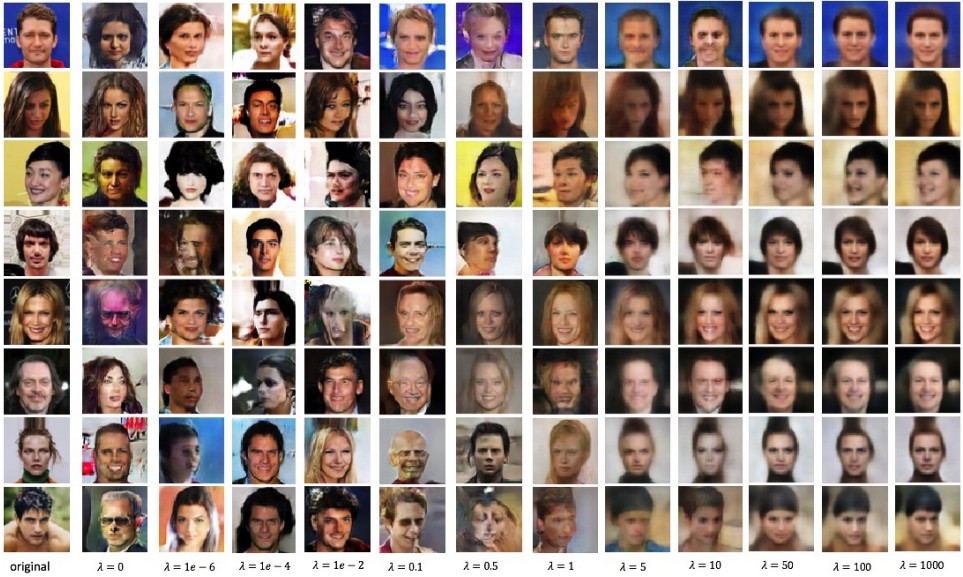

Figure 7: ALICE's CelebA reconstructions with different hyper-parameter $\lambda$ (reported in Pu et al. (2017)). The first column is the original samples from test set, and the following columns are reconstructions corresponding to different values of $\lambda$.

ALICE uses hyper-parameter $\lambda$ to control the strength of the conditional entropy (CE) regularization in the objective function. When $\lambda$ is 0, ALICE's objective function reduces to ALI's. As observed from Fig 7, ALICE fails to reconstruct original images when hyper-parameter $\lambda$ is relative small. This is related to the non-identifiability issue of ALI as described in Li et al. (2017). As $\lambda$ increases, more patterns are captured, but the reconstructed images becomes more like the results of VAE – the images are all blurred. This is because ALICE replaces the intractable CE with its upper

bound cycle-consistency loss, which becomes $l_k$ norms under parametric assumption on $p(\mathbf{x}|\mathbf{z})$. So just like VAE-GAN hybrids, ALICE can be treated as a VAE-ALI hybrids and also suffer from similar drawbacks. Therefore, adding conditional entropy into ALI's objective function heuristically manifests a compromise of the strengths and weaknesses of ALI and VAE.

In contrast, Figure 6 shows that our proposed model AIM can overcome the limitations of ALI and VAE. AIM's reconstructions capture the identifiable features in the original images, and are also very sharp without blurriness. In fact, we observe that AIM is able to reconstruct the images' background (color, texture), which are always lost in ALICE's reconstructions for any choice of $\lambda$.

## A.2  ANALYSIS OF RECONSTRUCTION ERRORS

We present the change of mean squared errors (MSE) on the first 50 training epochs in Figure 8. As can be seen, on all datasets, the MSEs decrease significantly during the first 10 epochs and then stay steadily at the similar minimum levels. Although the scales of reconstruction errors on different datasets at the beginning are different because of the various complexity of data distributions, the converged values of reconstruction errors are very similar. This observation proves that our inference mechanism can efficiently and consistently infer the latent codes of data samples, even when the data distribution is complicated.

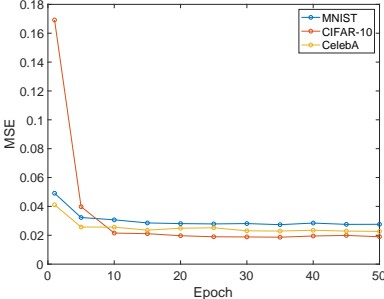

Figure 8: The MSE changes with epochs.

## A.3  UNSUPERVISED CONDITIONALLY GENERATED IMAGES

In Figure 9, we present (unsupervised) conditionally generated samples of MNIST and CelebA experiments. Each element in the first column stands for an original data $\mathbf{x}^{(i)}$. Then based on $\mathbf{x}^{(i)}$, we conditionally generate more images by sampling $\mathbf{z}$ from the posterior $q(\mathbf{z}|\mathbf{x}^{(i)})$. The conditionally generated images are shown in the same row of $\mathbf{x}^{(i)}$.

## A.4  RANDOMLY GENERATED IMAGES

In Figure 10, we show randomly generated images by sampling $\mathbf{z}$ from the marginal $p(\mathbf{z})$. The generation results can be easily improved by replacing DCGAN structure with more advanced GAN structures.

## B  HYPERPARAMETERS FOR MNIST

The details for the networks used in MNIST experiment are presented in Table 3. Here the column "feature maps" for the fully-connected (FC) layers represents the dimension of their output. FC $(\mu)$

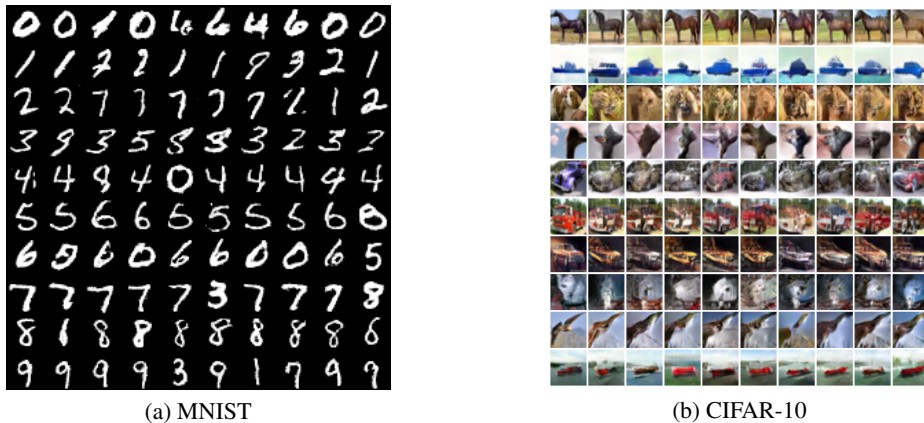

(a) MNIST            (b) CIFAR-10

Figure 9: Conditionally generated samples (unsupervised). The first column consists of original images.

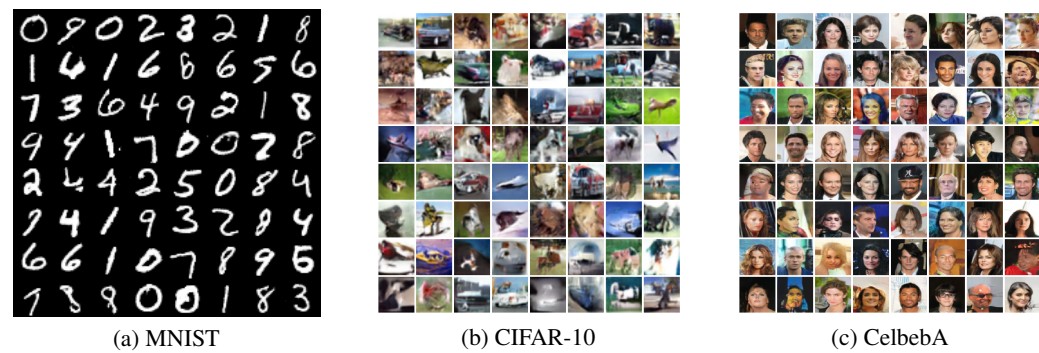

(a) MNIST      (b) CIFAR-10      (c) CelbebA

Figure 10: Randomly generated samples.

and FC ($\sigma$) are two fully-connected layers built on the previous common layer to separately predict mean $\mu(\mathbf{x})$ and variance $\sigma^2(\mathbf{x})$ of the conditional distribution $q(\mathbf{z}|\mathbf{x})$.

## C  HYPERPARAMETERS FOR CIFAR-10 AND CELEBA

The details for the networks used in CIFAR-10 and CelebA experiments are presented in Table 4. In order to reuse the same architecture for both datasets, we scaled CIFAR-10 images and center cropped CelebA images so that they have the same image size $64 \times 64$.

| Operation | Kernel | Strides | Feature maps | BN? | Dropout | Nonlinearity |
|---|---|---|---|---|---|---|
| $G(z) - 32 \times 1 \times 1$ input | | | | | | |
| FC | – | – | 1024 | $\checkmark$ | 0.0 | ReLU |
| FC | – | – | 6272 | $\checkmark$ | 0.0 | ReLU |
| Reshape to $128 \times 7 \times 7$ | | | | | | |
| Transposed convolution | $4 \times 4$ | $2 \times 2$ | 64 | $\checkmark$ | 0.0 | ReLU |
| Transposed convolution | $4 \times 4$ | $2 \times 2$ | 3 | $\times$ | 0.0 | Sigmoid |
| $D_{feature}(x) - 1 \times 28 \times 28$ input | | | | | | |
| Convolution | $4 \times 4$ | $2 \times 2$ | 64 | $\times$ | 0.0 | Leaky ReLU |
| Convolution | $4 \times 4$ | $2 \times 2$ | 128 | $\checkmark$ | 0.0 | Leaky ReLU |
| $D_{con}(D_{feature}(x)) - 128 \times 7 \times 7$ input | | | | | | |
| Reshape to $6272 \times 1 \times 1$ | | | | | | |
| FC | – | – | 1024 | $\checkmark$ | 0.0 | Leaky ReLU |
| FC | – | – | 64 | $\times$ | 0.0 | Linear |
| FC | – | – | 1 | $\times$ | 0.0 | Sigmoid |
| $E(D_{feature}(x)) - 128 \times 7 \times 7$ input | | | | | | |
| Convolution | $4 \times 4$ | $2 \times 2$ | 64 | $\checkmark$ | 0.0 | Leaky ReLU |
| Reshape to $576 \times 1 \times 1$ | | | | | | |
| FC($\mu$) | | | | | | |
| FC | – | – | 32 | $\checkmark$ | 0.0 | Leaky ReLU |
| FC | – | – | 32 | $\times$ | 0.0 | Linear |
| FC($\sigma$) | | | | | | |
| FC | – | – | 32 | $\checkmark$ | 0.0 | Leaky ReLU |
| FC | – | – | 32 | $\times$ | 0.0 | Linear |
| Optimizer | Adam ($\alpha = 2 \times 10^{-4}$, $\beta_1 = 0.5$) | | | | | |
| Learning rate decay | $\alpha = \alpha/\sqrt{t}, t = epoch$ | | | | | |
| Batch size | 100 | | | | | |
| Epochs | 400 | | | | | |
| Leaky ReLU slope | 0.1 | | | | | |
| Weight, bias initialization | Isotropic gaussian ($\mu = 0$, $\sigma = 0.02$), Constant(0) | | | | | |

Table 3: MNIST model hyperparameters.

| Operation | Kernel | Strides | Feature maps | BN? | Dropout | Nonlinearity |
|---|---|---|---|---|---|---|
| $G(z) - 64 \times 1 \times 1$ input | | | | | | |
| Transposed convolution | $4 \times 4$ | $1 \times 1$ | 1024 | √ | 0.0 | ReLU |
| Transposed convolution | $4 \times 4$ | $2 \times 2$ | 512 | √ | 0.0 | ReLU |
| Transposed convolution | $4 \times 4$ | $2 \times 2$ | 256 | √ | 0.0 | ReLU |
| Transposed convolution | $4 \times 4$ | $2 \times 2$ | 128 | √ | 0.0 | ReLU |
| Transposed convolution | $4 \times 4$ | $2 \times 2$ | 3 | × | 0.0 | Tanh |
| $D_{feature}(x) - 3 \times 64 \times 64$ input | | | | | | |
| Convolution | $4 \times 4$ | $2 \times 2$ | 128 | × | 0.0 | Leaky ReLU |
| Convolution | $4 \times 4$ | $2 \times 2$ | 256 | √ | 0.0 | Leaky ReLU |
| Convolution | $4 \times 4$ | $2 \times 2$ | 512 | √ | 0.0 | Leaky ReLU |
| Convolution | $4 \times 4$ | $2 \times 2$ | 1024 | √ | 0.0 | Leaky ReLU |
| $D_{con}(D_{feature}(x)) - 1024 \times 4 \times 4$ input | | | | | | |
| Convolution | $4 \times 4$ | $1 \times 1$ | 1 | × | 0.0 | Sigmoid |
| $E(D_{feature}(x)) - 1024 \times 4 \times 4$ input | | | | | | |
| Convolution | $4 \times 4$ | $2 \times 2$ | 512 | × | 0.0 | Leaky ReLU |
| Convolution | $4 \times 4$ | $2 \times 2$ | 512 | √ | 0.0 | Leaky ReLU |
| Convolution | $2 \times 2$ | $1 \times 1$ | 256 | √ | 0.0 | Leaky ReLU |
| Convolution | $1 \times 1$ | $1 \times 1$ | 128 | √ | 0.0 | Leaky ReLU |
| Reshape to $512 \times 1 \times 1$ | | | | | | |
| FC($\mu$) | | | | | | |
| FC | – | – | 64 | √ | 0.0 | Leaky ReLU |
| FC | – | – | 64 | × | 0.0 | Linear |
| FC($\sigma$) | | | | | | |
| FC | – | – | 64 | √ | 0.0 | Leaky ReLU |
| FC | – | – | 64 | × | 0.0 | Linear |

| | |
|---|---|
| Optimizer | Adam ($\alpha = 2 \times 10^{-4}$, $\beta_1 = 0.5$) |
| Learning rate decay | $\alpha = \alpha/\sqrt{t}$, $t = epoch$ |
| Batch size | 100 |
| Epochs | 300 |
| Leaky ReLU slope | 0.2 |
| Weight, bias initialization | Isotropic gaussian ($\mu = 0$, $\sigma = 0.02$), Constant(0) |

Table 4: CIFAR-10 and CelebA model hyperparameters.

