# OpenReview forum: "AIM: Adversarial Inference by Matching Priors and Conditionals"
_ICLR.cc/2019/Conference_

### Official Review · AnonReviewer2 · 2018-11-03
**Interesting idea, but needs more work**

**Rating:** 6
**Confidence:** 4

**Review:**

UPDATE (after author response):

Thank you for updating the paper, the revised version looks better and the reviewers addressed some of my concerns. I increased my score.

There's one point that the reviewers didn't clearly address:  "It might be worth evaluating the usefulness of the method on higher-dimensional examples where the analytic forms of q(x|z) and q(z) are known, e.g. plot KL between true and estimated distributions as a function of the number of dimensions." Please consider adding such an experiment.

The current experiments show that the method works better on low-dimensional datasets, but the method does not seem to be clearly better on more challenging higher dimensional datasets.  I agree with Reviewer1 that "Perhaps more ambitious applications would really show off the power of the model and make it standout from the existing crowd." Showing that the method outperforms other methods would definitely strengthen the paper.

Section 5.4: I meant error bars in the numbers in the text, e.g. 13 +/- 5.

---------

The paper proposes a new loss for training deep latent variable models. The novelty seems a bit limited, and the proposed method does not consistently seem to outperform existing methods in the experiments. I'd encourage the authors to add more experiments (see below for suggestions) and resubmit to a different venue.

Section 4:
- q(z) seems to be undefined. Is it the aggregated posterior?
- How is equation (1) related to ELBO that is used for training VAEs?

Some relevant references are missing: I’d love to see a discussion of how this loss relates to other VAE-GAN hybrids.

VEEGAN: Reducing mode collapse in GANs using implicit variational learning
https://arxiv.org/pdf/1705.07761.pdf

Distribution Matching in Variational Inference
https://arxiv.org/pdf/1802.06847.pdf


Section 5.1:
- The quantitative comparison measures MSE in pixel space and inception score, neither of which are particularly good measures for measuring the quality of how well the conditionals match. I’d encourage the authors to consider other metrics such as log-likelihood.

- It might be worth evaluating the usefulness of the method on higher-dimensional examples where the analytic forms of q(x|z) and q(z) are known, e.g. plot KL between true and estimated distributions as a function of the number of dimensions.

Section 5.4:
- The error bars seem quite high. Is there a reason why the method cannot reliably reduce mode collapse?

Minor issues:
- CIFAT-10 -> CIFAR-10

---

> ### Author Response · Authors · 2018-11-19
> **Author Response to AnonReviewer2**
>
> We thank Reviewer 2 for the constructive feedback. Here is our point-to-point response to the comments and questions raised in this review:
>
> 1. Section 4:
> - q(z) seems to be undefined. Is it the aggregated posterior?
>
> We are sorry for the confusion. Yes, q(z) is the aggregated posterior. In this paper, p() stands for the distribution on the generator, and q() stands for the distribution on the encoder.
>
> - How is equation (1) related to ELBO that is used for training VAEs?
>
> To better explain the relation between (1) and VAE, we have added a new Section 4.3.
>
> To summarize, equation (1) is our method’s objective. It means that AIM performs marginal distribution matching in the latent space and conditional distribution matching in the data space. But this objective cannot be optimized directly, so we transfer the problem using (3). It turns out VAE can be derived in a similar manner. Specifically, the objective of VAE can be understood as the equation (5), which is like the “reverse version” of equation (3).  By reverse, we mean that VAE can be explained as performing marginal distribution matching in the data space and conditional distribution matching in the latent space. Note that the RHS of (5) is the well-known VAE form, i.e. a regularization on z (I_vae) plus a reconstruction term on x (II_vae). But we actually get it from a perspective different from ELBO. ELBO is a lower bound of the log-likelihood of the data, and we maximize ELBO in order to maximize the log-likelihood. However, in equation (5), we do not have any inequality and are not directly trying to increase the likelihood. Instead, the LHS of (5) is the summation of KL-divergence between the conditionals on z and between the marginals on x. This is the quantity that VAE tries to minimize, from our perspective motivated by (3).
>
> 2. Some relevant references are missing: I’d love to see a discussion of how this loss relates to other VAE-GAN hybrids.
>
> Thank you for bringing these work to our attention. We have cited and discussed them in our updated draft. We also added another adversarial inference paper (adversarial variational Bayes). They are discussed in the second and fourth paragraph in the Related Work section. Empirical comparison with VEEGAN has also been added to Section 5.3.
>
> 3. Section 5.1
>
> We use MSE as the measure of how well our model reconstructs the samples. While there may not exist an absolutely perfect measure, we think the relative improvement on one measure still provides lots of information. For example, the best baseline model in Section 5.1 has MSE 0.080 on MNIST, while our model has only 0.026. On CIFAR-10, the best baseline MSE is 0.416, while ours is only 0.019. Moreover, all these improvements on MSE do not come with any compromise on generation. In fact, our model even improves the generating performance over GAN with the same architecture.
>
> But per the reviewer’s request, we add another measure called “percentage of high-quality samples” to our mode-collapse experiment, motivated by the experiments in VEEGAN. The results are summarized in Table 2. We observe that the best baseline model covers about 24.6 modes with 40% generated samples to be of high quality, while our model can cover all the 25 modes with more than 80% high-quality samples. This together with the inception score provides strong evidence that AIM can generate higher-quality samples.
>
> 4. Section 5.4
>
> We are not very sure which error bar was the reviewer referring to. But for better illustration, we have summarized the results in a table (Table 2). And from that, we can see that our model covers all of the 25 modes every time with high-quality samples, and can indeed reliably reduce the mode collapse.
>
> 5. Minor issue
>
> Thank you for pointing out this typo =) We have corrected it!

---

> ### Author Response · Authors · 2018-12-14
> **Second-round response to Reviewer 2**
>
> Dear Reviewer 2,
>
> We really appreciate that you considered our updates of the paper and increased the score!
>
> In the new response below, we have designed a high-dimensional experient and compared our method AIM with ALI, VEEGAN, and VAE.
> -------------------------------------------------------------------------------------------------------------------------------------------------------------------
>
> Experiment settings:
> The latent code z is of dimension 16, and the data X is of dimension 256. The relation between z and X is X = Az + N(0, 0.01^2 * I), where A is a Gaussian matrix with i.i.d entries from N(0, 0.05^2). Note that A was generated at first and then fixed thereafter. We then randomly sampled 200,000 z’s from the standard Normal distribution (16-dim), and then mapped them to the data space using A. Then we chose 100,000 of the samples for training, 50,000 for validation, and 50,000 for testing. The results are summarized below. The KL-divergence is calculated using the ITE-package [1] following the Adversarial Variational Bayes [2] paper.
>
>
>                                first epoch                               best KL                   best epoch/total epochs
>                          z                         X                     z                     X
> ---------------------------------------------------------------------------------------------------------------------------
> AIM                0.04                   76.54               0.31               2.02                    31 / 100
> ---------------------------------------------------------------------------------------------------------------------------
> ALI                  0.07                   62.83              0.37               5.66                    30 / 100
> ---------------------------------------------------------------------------------------------------------------------------
> VEEGAN         0.07                   47.95              0.33             11.41                    36 / 100
> ---------------------------------------------------------------------------------------------------------------------------
> VAE                 2.16                 619.30              0.02               9.15                    92 / 100
> ---------------------------------------------------------------------------------------------------------------------------
>
> In the "first epoch" column, we report the KL-divergence of the first epoch. We choose the "best epoch" to be the epoch when KL(z, z_fake) + KL(x, x_fake) attains its minimum on the validation set, and then report the KL-divergence of the "best epoch" on the test set.
>
> From the table, we observe that VAE achieves a much smaller KL on z. One important reason could be its explicit minimization of KL(z, z_fake). AIM and the other two adversarial methods have similar z-space KL-divergence.
>
> In the x-space, AIM has the best result, followed by ALI. VEEGAN has a slight improvement on z-space KL, but sacrifices the performance on x. Since in this case X only has one mode, our hypothesis is that VEEGAN makes a trade-off between alleviating mode collapse and maintaining the generating quality within each mode. However, this needs more experiments to confirm. VAE does not perform well on x-space, and its convergence seems much slower than the other methods. The KL on x may continue decreasing if we run other hundreds of epochs. But within 100 epochs, it does not give competitive results even though the experiment settings are very close to the VAE assumptions. However, our X does have covariance between different dimensions, while VAE assumes independent features to use the L2 loss. From our experiment, this violation indeed has a large impact on the performance.
>
> Overall, the experiment results are on par with our reports in the paper. AIM not only effectively infers the code z, but also uses the inference mechanism to further improve generating quality.
>
> References:
> [1] Szabo, Zoltan. "Information theoretical estimators (ite) toolbox." 2013.
> [2] Mescheder, Lars, Sebastian Nowozin, and Andreas Geiger. "Adversarial Variational Bayes: Unifying variational autoencoders and generative adversarial networks." ICML, 2017.

---

### Official Review · AnonReviewer1 · 2018-11-05
**Ok paper, some nice comparisons, but too similar to existing models**

**Rating:** 4
**Confidence:** 5

**Review:**

This paper presents a variant of the adversarial generative modeling
framework, allowing it to incorporate an inference mechanism. As such it is
very much in the same spirit as existing methods such as ALI/BiGAN. The
authors go through an information theoretic motivation but end up with the
standard GAN objective function plus a latent space (z) reconstruction
term. The z-space reconstruction is accomplished by first sampling z from
its standard normal prior and pushing that sample through the generator to
get sample in the data space (x), then x is propagated through an encoder
to get a new latent-space sample z'. Reconstruction is done to reduce the
error between z' and z.

Novelty: The space of adversarially trained latent variable models has
grown quite crowded in recent years. In light of the existing literature,
this paper's contribution can be seen as incremental, with relatively low novelty.

In the end, the training paradigm is basically the same as InfoGAN, with
the difference being that, in the proposed model,  all the latent
variables are inferred (in InfoGAN, only a subset of the latent
variables are inferred) . This difference was a design decision on the part of the InfoGAN
authors and, in my opinion, does not represent a significantly novel
contribution on the part of this paper.

Experiments: The experiments show that the proposed method is
better able to reconstruct examples than does ALI -- a result is not
necessarily surprising, but is interesting and worth further
investigation. I would like to understand better why it is that latent
variable (z) reconstruction gives rise to better x-space reconstruction.

I did not find the claims of better sample quality of AIM over ALI to be
well supported by the data. In this context, it is not entirely clear what
the significant difference in inception scores represents, though on this, the
results are consistent with those previously published

I really liked the experiment shown in Figure 4 (esp. 4b), it makes the
differences between AIM and ALI very clear. It shows that relative to ALI,
AIM sacrifices coherence between the "marginal" posterior (the distribution
of latent variables encoded from data samples) and the latent space
prior, in favor of superior reconstructions. AIM's choice of trade-off is
one that, in many contexts, one would happy to take as it ensures that
information about x is not lost -- as discussed elsewhere in the paper.
I view this aspect of the paper by far the most interesting.

Summary,
Overall, the proposed AIM model is interesting and shows promise, but I'm
not sure how much impact it will have in light of the existing literature
in this area. Perhaps more ambitious applications would really show off the
power of the model and make it standout from the existing crowd.

---

> ### Author Response · Authors · 2018-11-19
> **Author Response to AnonReviewer1**
>
> We thank reviewer 1 for the deep and insightful review. Here is our point-to-point response to the comments and questions raised in the review:
>
> 1. “The space of adversarially trained latent variable models has grown quite crowded in recent years.”
>
> Although there has been a large improvement in the topic of adversarial inference in recent years, some big issues are still not well addressed and limit the effectiveness of the inference mechanism in adversarial frameworks.
>
> Firstly, to the best of our knowledge, all of the works that attempt to incorporate the inference mechanism into GAN suffer from deteriorating the generation performance. This is supported by the paper [1], in which they conducted extensive experiments to compare many state-of-the-art models with DCGAN. The result shows that GAN variants with inference network perform worse than the standard DCGAN on image generation.
>
> Secondly, the inference performance is also very limited, and as the data distribution becomes more complicated, this issue will be more severe. For example, ALICE’s reconstruction performance on CIFAR-10 is much worse than that on MNIST.
>
> To the best of our knowledge, we are the first to successfully handle these two issues simultaneously. For generation performance, our model AIM does not deteriorate the generation performance but actually further improve it compared with GAN with the same architecture. For the inference, AIM consistently achieves better results on even complicated distributions.
>
> 2. “I would like to understand better why it is that latent variable (z) reconstruction gives rise to better x-space reconstruction.”
>
> We have added a new Section 4.3 to demonstrate the connection between our model and VAE. Specifically, the objective of VAE can be understood as the equation (5), which is like the “reverse version” of equation (3).  By "reverse", we mean that VAE can be explained as performing marginal distribution matching in the data space and conditional distribution matching in the latent space, while our model performs marginal distribution matching in the latent space and conditional distribution matching in the data space.
>
> Note that the latent z reconstruction alone does not guarantee a better data space reconstruction, just like a stand-alone x reconstruction of VAE will not work without the help of regularization on z. Our method has two simultaneous conditions on the generator, encoder, and discriminator: the generator has to generate samples that can fool the discriminator and the encoder has to bring these generated samples back to their latent codes. So the generator needs to not only generate samples that look real, but also map the latent codes to the “correct” locations (e.g. modes). Otherwise, the encoder will have a hard time to map the samples back (more precisely in our case, will have a low likelihood).
>
> Another interesting property of (5) is that it actually provides a new perspective on VAE’s objective, different from the maximum likelihood point of view. Note that there is also no inequality in (5), unlike the ELBO approach. We can get VAE’s objective by decomposing the summation of the KL-divergence between posteriors on z and between marginals on x.
>
> 3. I did not find the claims of better sample quality of AIM over ALI to be well supported by the data. In this context, it is not entirely clear what the significant difference in inception scores represents.
>
> Probably the 2D Gaussian mixture result will provide an insight here. From Table 2, we can see that ALI’s generated samples only cover on average 16 (out of 25) modes while our method’s can cover 25 every time. The ALI’s result we show in Figure 3 is the best-covering result they report, and we include it only to give more insights on the difference between our method and the joint distribution matching scheme.
>
> From the quantitative results in Table 1, we also observe that AIM gives a higher inception score than ALI, and this happens when AIM also has a much lower reconstruction error. The main takeaway message is that, compared to the joint distribution matching of ALI, the separate marginal and conditional matching of AIM leads to better reconstruction and generation. Actually, from Figure 2, the reconstructions of ALI are not always faithful even on the MNIST dataset. We think this is because that it is still very hard for the adversarial game to discover the dependency relation between x and z.
>
> Reference:
> [1] Distribution matching in variational inference.

---

> ### Author Response · Authors · 2018-12-18
> **Discussion with Reviewer 1**
>
> Dear Reviewer 1,
>
> We have added another high-dimensional experiment in which we compared the KL-divergence achieved by different models. The result has been posted in another reply.
>
> In the previous update, we have tried to address some of your concerns. Furthermore, we have added another section 4.3 to explain the connection between our method and VAE.
>
> Do you have any further suggestion? We would be glad to have more discussions with you!
>
> Thank you!

---

### Official Review · AnonReviewer3 · 2018-11-05
**A Review on Adversarial Inference by Matching Priors and Conditionals**

**Rating:** 7
**Confidence:** 4

**Review:**

The goal this is work is to develop a generative model that enjoys the strengths of both GAN and VAE without their inherent weaknesses. The paper proposes a learning framework, in which a generating process p is modeled by a neural network called generator, and an inference process q by another neural network encoder. The ultimate goal is to match the joint distributions, p(x, z) and q(x, z), and this is done by attempting to match the priors  p(z) and q(z) and matching the conditionals p(x|z) and q(x|z). As both q(z) and q(x|z) are impossible to sample from, the authors mathematically expand this objective criterion and rewrite to be dependent only on p(x|z), q(x) and q(z|x), that can be easily sampled from. In the main part of the work, the authors use the f-divergence theory (Nowozin et al., 2016) to present the optimization problem as minmax optimization problem, that is learned using an adversarial game, using training and inference algorithms that are proposed by the authors. In experiments, the authors consider both reconstruction and generation tasks using the MNIST, CIFAR10 and CelebA datasets. Results show that the proposed method yields better MSE reconstruction error as better as a higher inception scores for the generated examples, compared to a standard GAN and a few other methods.

This work establishes an important bridge between the VAE and GAN framework, and has a a good combination of theoretical and experimental aspects. Experiments results are encouraging, even though only relatively simple and small datasets were used. Overall, I would recommend accepting the paper for presentation in the conference.

---

> ### Author Response · Authors · 2018-11-19
> **Author Response to AnonReviewer3**
>
> We thank Reviewer 3 for the encouraging feedback and the precise summary of our work.
>
> For a fair comparison, we only conduct experiments on the same datasets used in the related papers. But the architecture of our model (especially the generator and discriminator) can be easily replaced by more advanced state-of-the-art GANs for larger and more complicated datasets.
>
> FYI, we have added another Section 4.3 to explain the interesting relation between our method and VAE. And we have also added more experiments to Section 5.3.

---

### Meta-Review · Area_Chair1 · 2018-12-14

**Confidence:** 4
**Recommendation:** Reject

**Metareview:**

The paper proposes a method that aims to combine the strenghts of VAEs and GANs.

The paper establishes an interesting bridge between GANs and VAEs. The experimental results are encouraging, even though only relatively small datasets were used. It is encouraging that the method results in better reconstructions then ALI, a related method.

Some reviewers think that the paper contains limited novelty compared to the wealth of recent work on this topic (e.g. ALI/BiGAN). The paper's contribution is seen as incremental; e.g. the training is very similar to InfoGAN. Also, the claims of better sample quality over ALI seem insufficiently supported by the data.